# Structural insights into the mechanism of the membrane integral N-acyltransferase step in bacterial lipoprotein synthesis

Maciej Wiktor[1,*,†], Dietmar Weichert[1,*], Nicole Howe[1,*], Chia-Ying Huang[1,2], Vincent Olieric[2], Coilín Boland[1], Jonathan Bailey[1], Lutz Vogeley[1], Phillip J. Stansfeld[3], Nienke Buddelmeijer[4], Meitian Wang[2] & Martin Caffrey[1]

Lipoproteins serve essential roles in the bacterial cell envelope. The posttranslational modification pathway leading to lipoprotein synthesis involves three enzymes. All are potential targets for the development of new antibiotics. Here we report the crystal structure of the last enzyme in the pathway, apolipoprotein N-acyltransferase, Lnt, responsible for adding a third acyl chain to the lipoprotein's invariant diacylated N-terminal cysteine. Structures of Lnt from *Pseudomonas aeruginosa* and *Escherichia coli* have been solved; they are remarkably similar. Both consist of a membrane domain on which sits a globular periplasmic domain. The active site resides above the membrane interface where the domains meet facing into the periplasm. The structures are consistent with the proposed ping-pong reaction mechanism and suggest plausible routes by which substrates and products enter and leave the active site. While Lnt may present challenges for antibiotic development, the structures described should facilitate design of therapeutics with reduced off-target effects.

[1] Membrane Structural and Functional Biology (MS&FB) Group, School of Medicine and School of Biochemistry and Immunology, Trinity College Dublin, Dublin 2, Ireland. [2] Swiss Light Source, Paul Scherrer Institute, CH-5232 Villigen, Switzerland. [3] Department of Biochemistry, University of Oxford, South Parks Road, Oxford OX1 3QU, UK. [4] Institut Pasteur, Department of Microbiology, Biology and Genetics of the Bacterial Cell Wall Unit, Inserm Group Avenir, 25-28 Rue du Docteur Roux, 75724 Paris cedex 15, France. * These authors contributed equally to this work. † Present address: Laboratory of Biochemistry, Faculty of Biotechnology, University of Wroclaw, 50-383 Wroclaw, Poland. Correspondence and requests for materials should be addressed to M.C. (email: martin.caffrey@tcd.ie).

Lipoproteins are essential components of bacterial membranes. Lipoproteins' functions are myriad[1–5]. Some have enzymatic activity, as in the case of the β-lactamase (penicillinase) lipoprotein[6]. Others are enzyme activators[7] and inhibitors[8]. Many lipoproteins are components of complexes. A noted example is the tetra-heme cytochrome subunit of the photosynthetic reaction center[9]. Other complexes where lipoproteins play an important role include the β-barrel assembly machinery (BAM) where four of the five BAM subunits are lipoproteins[10,11]. Lipoproteins are also virulence factors against which mammals have evolved immune response systems[12,13]. These assorted functions make lipoproteins interesting, important and relevant biomacromolecules to understand. The current study focusses on the final step in the lipoprotein synthesis and maturation pathway.

Lipoproteins have an N-terminal cysteine residue where one or both of its functional groups is posttranslationally lipid modified. A diacylglyceryl moiety is attached by a thioether linkage to the invariant cysteine producing a diacylated lipoprotein (Fig. 1). Combining this modification with an N-acylation of cysteine's free α-amino group generates a triacylated lipoprotein. In most cases, the lipid modification affixes the protein to the membrane with the protein part outside the membrane. Depending on growth conditions, a lipoprotein can be di- or triacylated suggesting that posttranslational modification (PTM) is tightly controlled.

Lipoprotein synthesis begins with the full-length pre-prolipoprotein entering the inner membrane via the TAT or Sec secretion systems (Fig. 1). It has a membrane anchoring N-terminal signal sequence 10 to 25 residues long that ends with a consensus lipobox sequence of form $L(A/V)^{-4}-L^{-3}-A(S)^{-2}-G(A)^{-1}-C^{+1}$. The first step in the three-step PTM pathway is catalysed by lipoprotein diacylglyceryl transferase, Lgt[14], which attaches a diacylglyceryl moiety from phosphatidylglycerol (PG) to the lipobox invariant cysteine by a thioether link[15]. The diacylglyceryl modified (dagylated) cysteine is hereafter designated Cys*. As a result of the reaction, the prolipoprotein product is secured in the membrane by the signal sequence and by the lipid modification. The second PTM step is catalysed by lipoprotein signal peptidase, LspA[16,17], which cleaves the signal sequence from the prolipoprotein to the N-side of Cys*. The apo-lipoprotein product remains anchored in the membrane by its diacylglyceryl tail. For some lipoproteins no further modification is needed. They remain and function in the diacylated state. The rest are acted on by the third and last enzyme in the PTM pathway, apolipoprotein N-acyltransferase, Lnt[18]. This transferase N-acylates the Cys* of the apo-lipoprotein using preferentially phosphatidylethanolamine (PE) as a lipid substrate and generating the triacylated lipoprotein product[2,18–21].

Crystal structures of the first two enzymes in the PTM pathway have been reported. Lgt from *E. coli* was captured in complex with PG and the inhibitor palmitic acid[22]. The structure of LspA from *P. aeruginosa* was solved with the antibiotic globomycin in the active site of this aspartyl peptidase[17]. Here we describe the structure of Lnt, the last enzyme in the PTM pathway, which offers valuable insights into its mechanism of action and antibiotic development. In *E. coli*, the enzyme has been shown to use the *sn*-1 chain of PE preferentially for lipoprotein N-acylation in a two-step reaction[20]. Lnt is a member of the nitrilase family of enzymes with two domains, a membrane domain (MD) and a periplasmic nitrilase-like domain (ND) harbouring the enzyme's active site[21,23].

Nitrilases comprise a superfamily of thiol enzymes found in animals, plants, fungi and in certain prokaryotes[23]. They perform nitrile and amide hydrolysis and the N-acylation (reverse amidolysis) of proteins. The nitrilase reaction takes place in a globular domain with a conserved αββα fold and a Glu/Lys/Cys catalytic triad. Seven branches of the superfamily have additional domains. Lnt, with its MD, is one such example. In *E. coli*, Glu267, Lys335 and Cys387 were identified as the catalytic triad residues and homology modelling, with known nitrilase

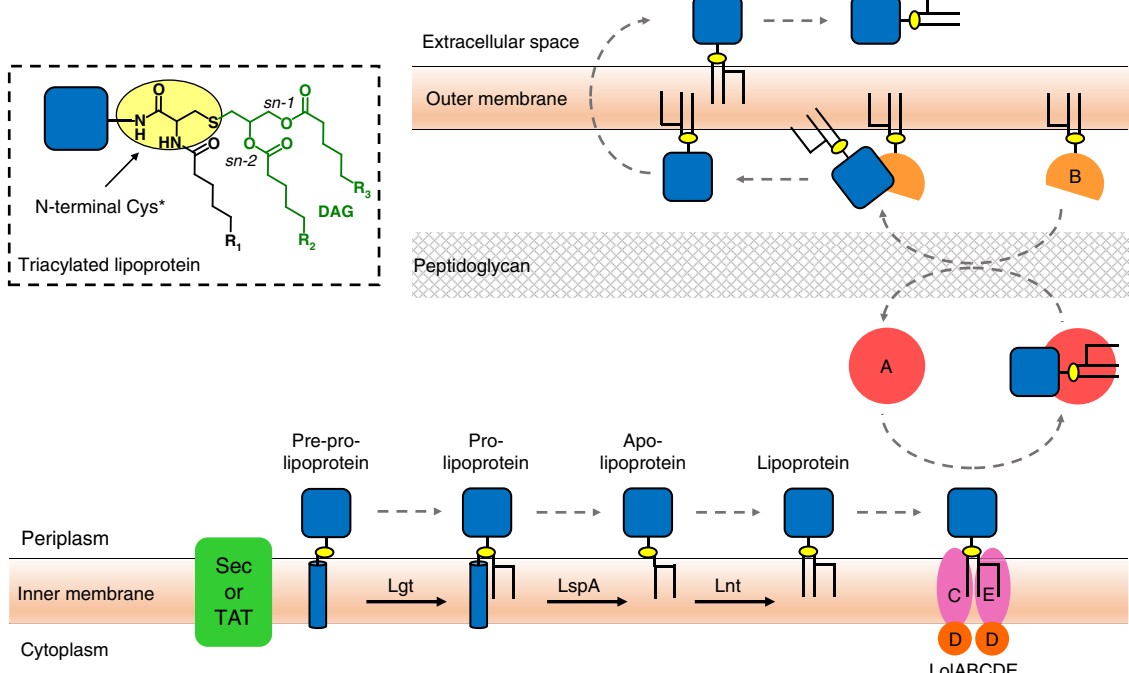

**Figure 1 | Lipoprotein posttranslational processing in Gram-negative bacteria.** The pathway includes the three processing enzymes Lgt, LspA and Lnt that reside in the inner membrane. Pre-prolipoproteins enter the membrane for processing via the Sec or TAT pathways. Trafficking to the outer membrane is by way of the Lol ABC transporter system. The inset shows the chemical structure of the triacylated N-terminal cysteine of a mature lipoprotein. DAG, diacylglyceryl.

structures as templates, provided a putative structure and a mechanism for Lnt action[24]. Clearly, a crystal structure of the full-length enzyme was required to rationalize these findings and conjectures and to provide a structural framework for understanding Lnt's mechanism of action.

## Results

**Structure determination.** A particular focus of this research group is on membrane proteins in *P. aeruginosa*. Accordingly, a structure of Lnt from this opportunistic human pathogen (LntPae) was selected for investigation. Because so much was known about Lnt from *E. coli* (LntEco), it too was included in the study. The two proteins have 39% sequence identity. The pure proteins were functionally active as demonstrated in assays where product formation was quantified with the synthetic biotinylated lipopeptide, fibroblast stimulating ligand-1 (FSL-1-biotin), as the second substrate (Fig. 2). Crystallization trials were undertaken using the lipid cubic phase method[25]. Crystals that diffracted to ~3 Å were obtained for both Lnt constructs. Seleno-methionine (Se-Met) labelling of LntEco for Se-single-wavelength anomalous diffraction (Se-SAD) phasing was used to solve the native LntEco structure to a resolution of 2.9 Å. The LntPae structure and that of an inactive Cys387Ala mutant of LntEco were solved by molecular replacement using the native LntEco structure as the search model. Data collection and refinement statistics are presented in Table 1. Representative electron density maps are shown in Supplementary Figs 2 and 3. The overall structure of the three constructs is similar. Co-evolutionary covariance analysis is consistent with the crystal structures[26] (Supplementary Fig. 4). For purposes of the discussion that follows the focus will be on the wild type LntEco construct.

**Overall architecture.** Lnt has two domains, a MD and a periplasmic ND with 230 and 278 residues, respectively (Fig. 3). The interfacial area between domains is 440 Å$^2$. The active site, defined by the catalytic triad, resides in the ND slightly above the membrane surface with an opening for substrates and products that leads into the bulk membrane. The MD consists of eight transmembrane helices (H1–H8) with both N and C termini in the cytoplasm (Fig. 4). The first six helices are arranged cylindrically with helices forming the wall of the cylinder and coiled counter-clockwise. At its periplasmic end, opposing helices across the cylinder's diameter are approximately equidistant giving the MD the appearance of an open ended cylinder (Fig. 4b). This periplasmic opening is covered by a highly conserved 27-residue long linker (L1) connecting H5 and H6 that includes two short helices, h1 and h2. The positioning of L1 across the top of the MD is due, in part, to Pro129 which introduces a kink in H5 causing it to bend away from H6 and to align with H4 (Fig. 4a). This enables conserved Arg139 on the periplasmic end of H5 to hydrogen bond with backbone carbonyls of highly conserved residues Phe146 and Trp148 in L1 thereby securing the loop as a periplasmic lid on the MD. The essential nature of Arg139 is borne out by mutational studies. A strong interaction between highly conserved Gly145 in L1 of the MD and Tyr388 (Fig. 5d) in the ND contributes to holding

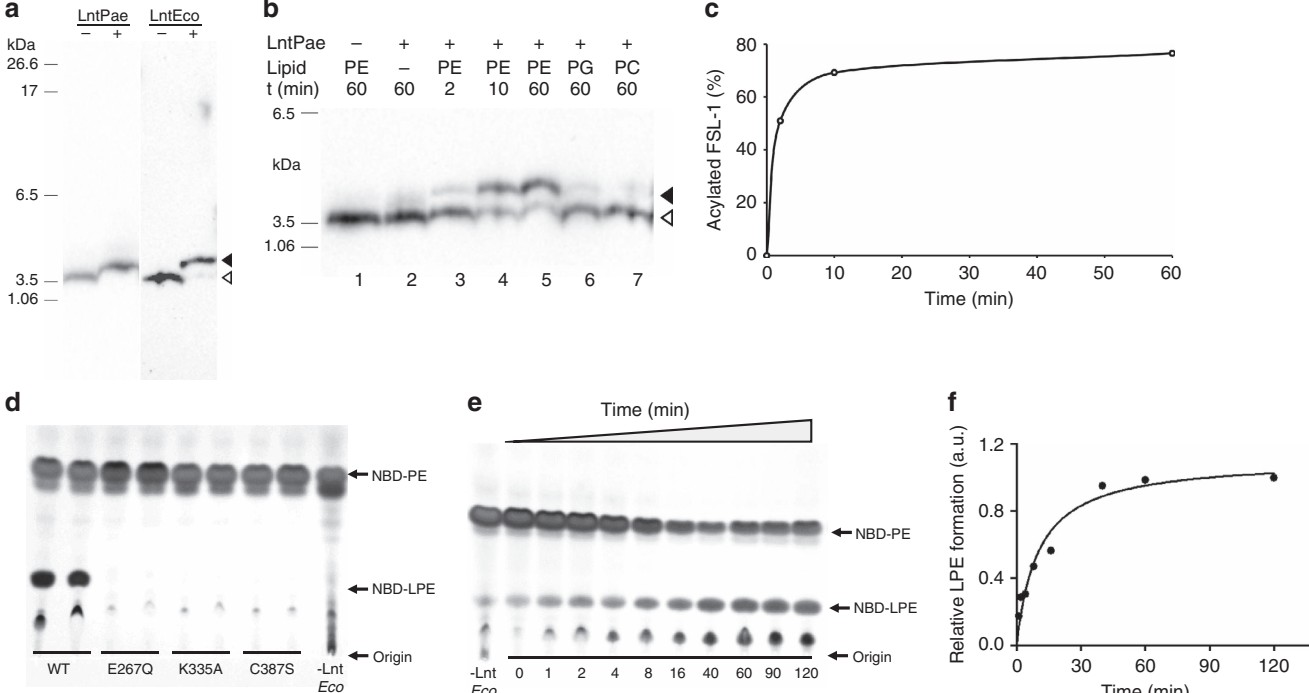

**Figure 2 | Lnt activity assay.** (**a**–**c**) Lnt activity monitored as a shift in the SDS-PAGE mobility of FSL-1-biotin on N-acylation with quantitation by western blotting. Uncropped images of the western blots including marker lanes are shown in Supplementary Fig. 1. (**d**,**e**) Lnt activity monitored by the conversion of NBD-PE to NBD-lyso-PE (NBD-LPE) by thin layer chromatography with quantitation by fluorescence. (**a**) N-acyl transferase activity of LntPae and LntEco is evident as a band shift towards higher molecular weight values resulting from a conversion of FSL-1-biotin (open arrow head) to N-acylated FSL-1-biotin (full arrow head) in the presence of a lipid donor. (**b**) Time course experiment and lipid head group specificity of LntPae. After a 1 h incubation at 37 °C in the presence of both substrates and enzyme (20 nM), the FSL-1-biotin band appeared ~3 mm higher in the gel than was observed for both negative control reactions without either DOPE or LntPae (lanes 1 and 2). The time dependence of the reaction is shown in lanes 3–5. When DOPE was replaced with DOPG or DOPC, the product was formed much less efficiently (lanes 6 and 7). (**c**) Densitometric analysis of the time-dependent data in **b**. (**d**) LntEco activity measurements for wild type (WT) and mutants E267Q, K335A and C387S. The reaction was stopped after 60 min. Data are shown for duplicate reaction measurements. (**e**) Time course of lyso-PE production catalysed by LntEco. (**f**) Densitometric analysis of the time-dependent data in **e**.

**Table 1 | Data collection and refinement statistics for Lnt.**

|  | Lnt*Eco*-Se* | Lnt*Eco* | Lnt*Eco*C387A | Lnt*Pae* |
|---|---|---|---|---|
| PDB code | — | 5N6H | 5N6L | 5N6M |
| *Data collection* |  |  |  |  |
| Space group | $P2_12_12_1$ | $P2_12_12_1$ | $P2_12_12_1$ | $C2$ |
| Cell dimensions |  |  |  |  |
| $a$, $b$, $c$ (Å) | 54.56, 142.44, 201.39 | 54.2, 142.3, 199.92 | 55.22, 142.95, 197.71 | 152.73, 39.23, 102.25 |
| $\alpha$, $\beta$, $\gamma$ (°) | 90, 90, 90 | 90, 90, 90 | 90, 90, 90 | 90, 116.26, 90 |
| Beamline | X10SA-PXII | X10SA-PXII | X10SA-PXII | X06SA-PXI |
| Wavelength (Å) | 0.9792 | 1.0 | 1.0 | 1.0 |
| No. of crystals | 3 | 3 | 3 | 2 |
| Total data (°) | 450 | 540 | 160 | 300 |
| Resolution (Å) | 50–3.97 (4.08–3.97) | 50–2.90 (3.00–2.90) | 50–2.90 (2.98–2.90) | 50–3.10 (3.20–3.10) |
| $R_{meas}$ | 0.29 (2.93) | 0.34 (3.00) | 0.24 (1.57) | 0.28 (1.53) |
| $I/\sigma_I$ | 8.97 (0.83) | 9.52 (1.00) | 7.72 (1.32) | 6.50 (1.43) |
| Completeness (%) | 96.0 (66.2) | 100 (100) | 97.9 (86.3) | 98.9 (99.2) |
| Multiplicity | 16.28 (10.91) | 19.67 (20.40) | 5.42 (3.55) | 5.40 (5.51) |
| $CC_{1/2}$ | 0.99 (0.26) | 0.99 (0.27) | 0.99 (0.42) | 0.98 (0.41) |
| $CC_{anom}$ | 0.3 | — | — | — |
| Mosaicity (°) | 0.21 | 0.24 | 0.31 | 0.25 |
| Phasing | Se-SAD | MR | MR | MR |
| Resolution range (Å) | 50–4.5 |  |  |  |
| Heavy atoms sites | 20 Se |  |  |  |
| Correlation coefficient (all/weak) | 49.40/18.80 |  |  |  |
| *Refinement* |  |  |  |  |
| Resolution (Å) |  | 50–2.90 | 50–2.90 | 50–3.10 |
| No. of reflections $R_{work}/R_{free}$ |  | 35266/1763 | 34807/1699 | 10157/1005 |
| $R_{work}/R_{free}$ |  | 0.22/0.25 | 0.23/0.26 | 0.22/0.27 |
| r.m.s. deviations |  |  |  |  |
| Bond lengths (Å) |  | 0.003 | 0.004 | 0.004 |
| Bond angles (°) |  | 0.728 | 0.798 | 0.855 |
| B-factor |  |  |  |  |
| Proteins chain A |  | 72.38 | 78.49 | 64.97 |
| Proteins chain B |  | 87.98 | 90.88 | — |
| Ligands |  |  |  |  |
| 9.9 MAG |  | 83.39 | 98.79 | 68.46 |
| Glycerol |  | 93.07 | 99.31 | 73.66 |
| Citrate |  | — | — | 86.67 |
| $H_2O$ |  | 72.61 | 74.83 | 48.17 |
| Ramachandran Plot |  |  |  |  |
| Favoured (%) |  | 96.48 | 97.68 | 95.26 |
| Allowed (%) |  | 3.52 | 2.32 | 4.74 |
| Outliers (%) |  | 0.00 | 0.00 | 0.00 |
| MolProbity Clash score |  | 6.26 | 9.94 | 8.52 |

PDB, protein data bank.
*Data processing statistics is reported with Friedel pairs separated. Values in parentheses are for the highest resolution shell.

the two domains together where they meet at the periplasmic surface of the membrane. This, in turn, helps position the catalytic Cys387 for reaction (Fig. 5). Cys387 in the ND sits above the center of the MD about 13 Å from where L1 crosses its cylindrical opening (Fig. 3a). Mapping conserved residues onto the structure identifies this region, between the two domains, as the putative active site pocket (Supplementary Figs 5a,6).

On the opposite, cytoplasmic end of the MD, the same cylindrical arrangement of helices H1–H6 holds with one exception. H2 is tilted into the core of the cylinder effectively closing this end of the protein (Fig. 4c). Arg123 on the cytoplasmic end of H5 interacts with Thr48 on H2 and with other local residues creating a water-tight seal between the two sides of the membrane. H7 and H8 in the MD are bridged by the ND in the periplasm. The connection between the two domains is by way of two long linkers (L2, L3) that look and possibly act like braces crossing over one another at the back of the ND (Fig. 3a, Supplementary Fig. 7a). The H7/H8 helix pair sits to one end of

the MD (hereafter referred to as the back end), run approximately parallel to one another, are oriented almost normal to the membrane plane and are separated from each other by the width of the MD. Arg438 on the ND has extensive interactions with Thr478 and Gly479 in the L3 brace to H8. Furthermore, highly conserved Thr481 on the L3 brace interacts with highly conserved Glu435 on the ND (Supplementary Fig. 7b). Collectively, these interactions presumably help poise the catalytic triad in the ND above the MD for reaction and contribute to Lnt's overall structural integrity and function. Interestingly, a Thr481Arg mutation inhibited the S-acylation step in the transferase reaction consistent with an uncoupling of the two domains[21].

Despite having a very high content of hydrophobic residues, H3 and H4 appear to extend above the membrane interface into the periplasm (Figs 3a and 4a, Supplementary Fig. 5b). H3 is bent at conserved Gly71 and coils over H4 in an anticlockwise direction. We speculate that the periplasmic surface of H3 and H4 forms one side of the portal for amphiphilic substrates to enter the

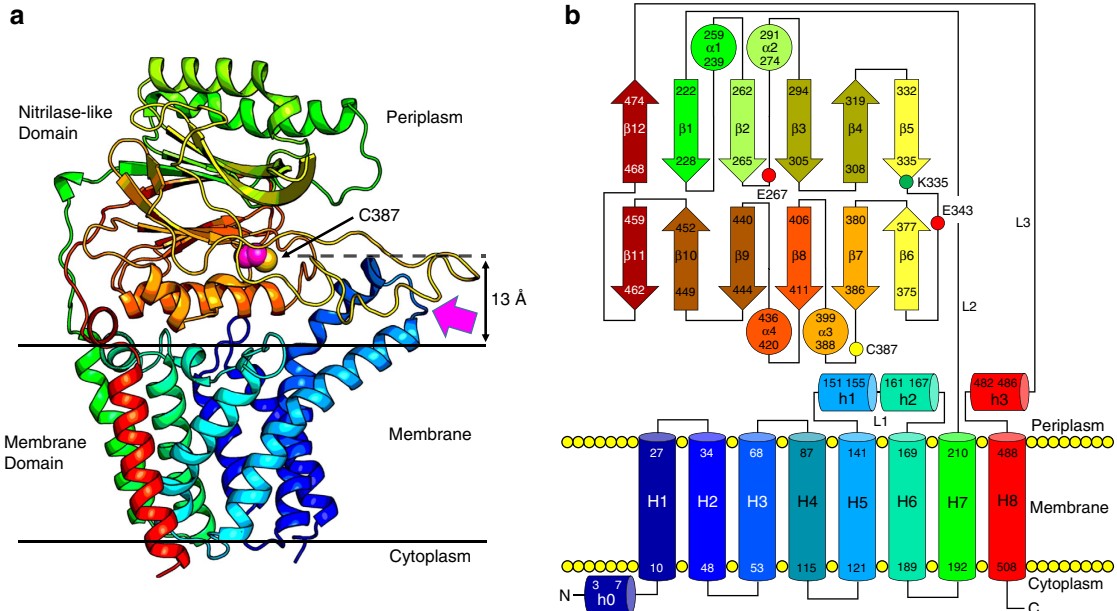

**Figure 3 | Overall architecture of Lnt from *E. coli*.** (**a**) View from the membrane plane. The protein has two domains, a membrane domain and a periplasmic nitrilase-like domain. The structure is shown in cartoon representation and rainbow colour coded from N (blue) to C terminus (red). The catalytic cysteine Cys387 side chain is shown in sphere representation (carbon, magenta; sulfur, yellow). The magenta arrow indicates the proposed substrate entry portal and identifies what is referred to as the front of the enzyme. Approximate location of the membrane boundaries are shown as horizontal lines. Cys387 sits ~13 Å above the bulk membrane surface. (**b**) Schematic representation of the secondary structure elements in the LntEco structure. Colour coding follows that used in **a**.

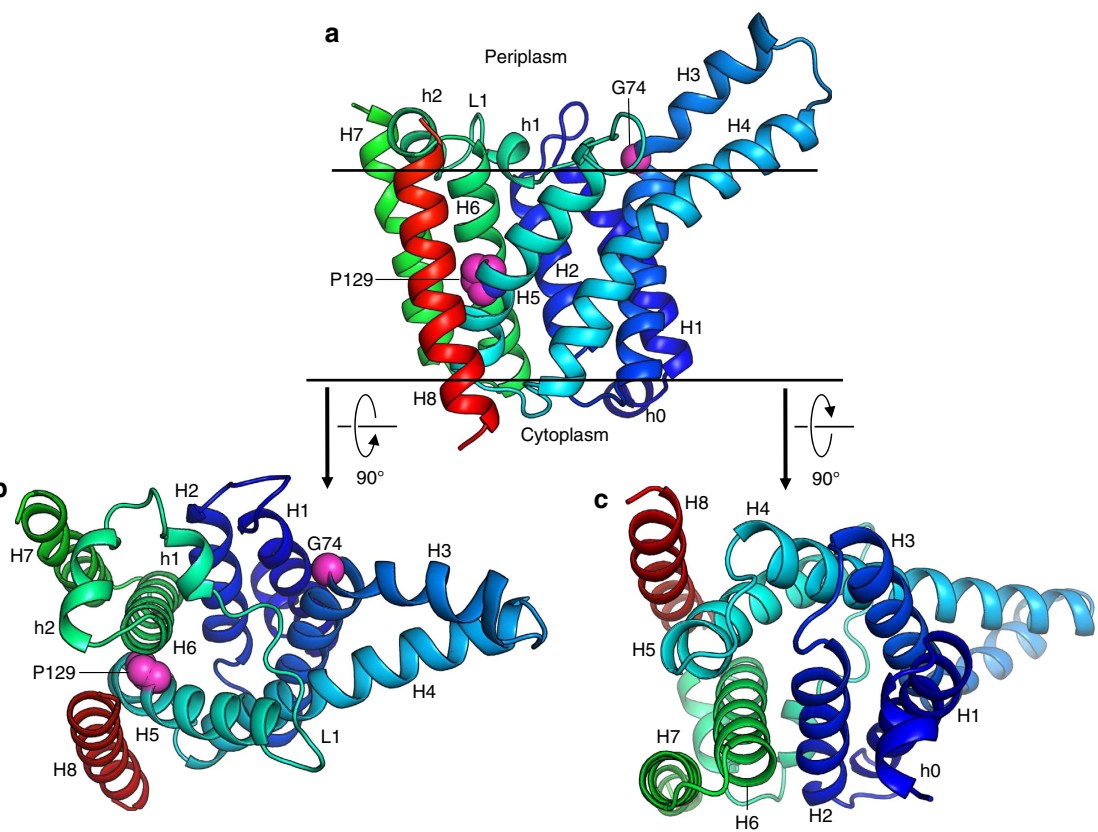

**Figure 4 | Membrane domain of LntEco.** (**a**) View from the membrane plane. (**b**) View from the periplasm. (**c**) View from the cytoplasm. Colour coding as in Fig. 3. Gly74 and Pro129 side chains shown as spheres.

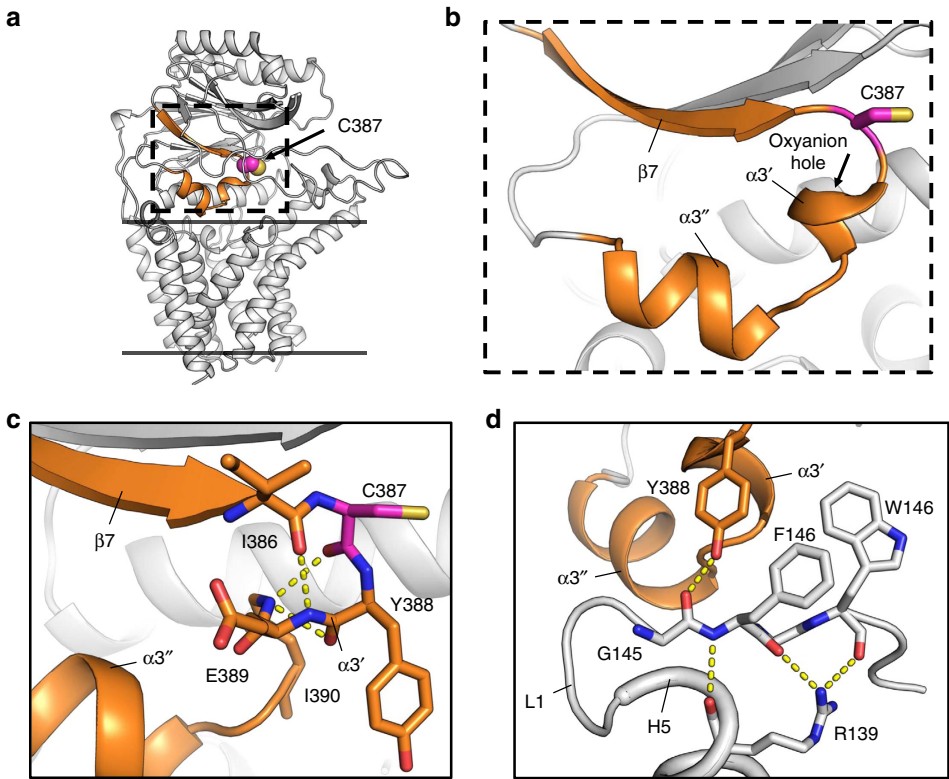

**Figure 5 | Nucleophilic elbow in LntEco.** (**a**) The nucleophilic elbow (orange) consisting of a β-strand-turn-helix (dashed box) shown in context of the overall Lnt structure (grey). (**b**) Expanded view of the boxed region in **a** showing the catalytic Cys387 in the turn. The oxyanion hole created by backbone amides in α3′ is indicated. (**c**) Coordination between residues in the turn and the α3′ helix. Dashed lines correspond to distances of ≤3.5 Å. (**d**) Residues in L1 are coordinated to the core of the MD via Arg139 in H5 and to the nucleophilic elbow via Tyr388.

active site. In addition to Gly71, H3 has three conserved glycines (Gly60, Gly64 and Gly66) in close proximity to one another. This suggests that H3 has considerable flexibility, possibly to accommodate and to orient into the active site the N-terminal diacylglyceryl–cysteine in differently sized and shaped lipoproteins. Crystallography and molecular dynamics simulations (MDS) data support this proposal (Supplementary Fig. 5c and d).

The ND is globular in shape and rests on the periplasmic surface of the MD (Figs 3a and 6). It emerges from the MD as a 277-long polypeptide connecting H7 and H8 (Supplementary Fig. 7). As noted, the two domains are linked by way of two brace-like linkers (L2, L3) at the back of the ND (Supplementary Fig. 7a). L2 and L3 connect H7 and H8 in the MD with the first (β1) and last β-strands (β12) in the ND, respectively. L3 includes a short helix (h4) towards the junction with H8. The ND is of the four-layer αββα sandwich fold type. Helices α1 and α2 form the upper layer that resides atop the protein in the periplasm with helices α3 and α4 forming the lower layer that sits on the periplasmic surface of the MD. α3 is really a pair of helices. The first is a single-turn helix (α3′) and is part of a β-strand-turn-helix motif termed a nucleophilic elbow in which resides the catalytic Cys387 (Figs 5 and 6). Helix α3′ includes highly conserved residues Tyr388 and Glu389 both of which were shown in mutational work to be essential[24]. The second helix in the pair (α3″) is two turns long. It is separated from the lower β-sheet of the ND and tilts into the core of the MD providing shape complementarity between the two domains. The two sheets of the nitrilase sandwich have different alternation of parallel and anti-parallel strands and, as usual, are twisted. The upper and lower αβ halves are connected on the front end of the ND by a 40 residue long loop with a short helix (α**) between β5 and β6 and on the back end by the MD tethering linkages L2 and L3 between

β11 and β12. Layers within the sandwich are held together primarily by hydrophobic interactions. The back end of the domain consists of short loops and, as a result, it is relatively flat and featureless. By contrast, the front end includes several loops of varying length that extend away from the ND approximately parallel to the membrane plane arranged roughly in a ring around the catalytic triad. Together with the periplasmic H3 and H4 helical extensions, these loops create an opening to the membrane and a funnel-shaped pocket at the base of which sits the catalytic triad (Fig. 7a). They resemble arms that reach out from the active site creating a pocket for substrates to enter and for products to leave. For convenience of description these active site cavity surrounding loops are hereafter referred to as arms and are identified numerically clockwise around the catalytic Cys387 (Fig. 7a). Residues Pro353, Phe358 and Met362 on Arm3 and Trp237 on Arm7 have been identified in mutational studies as important or essential[21]. Highly conserved Gly447 resides at the end of Arm6. Given the location of these key residues and the fact that they do not interact notably with other parts of the protein we speculate that their essentiality arises from the critical roles the respective residues and the loops in which they reside play in guiding substrates and products into and out of the active site.

**A portal into and out of the active site pocket.** A cleft exists between H4 and H5 in the periplasmic leaflet of the MD that extends out of the bulk membrane into the space between Arm 1 and Arm 3 (Fig. 7a, Supplementary Fig. 5b). This long hydrophobic opening leads into the active site pocket. We propose that it serves as the portal through which lipid and lipoprotein substrates enter the active site. In support of this, the structures of Lnt solved in the course of this study contain a varying number of structured monoolein lipid molecules from the mesophase in

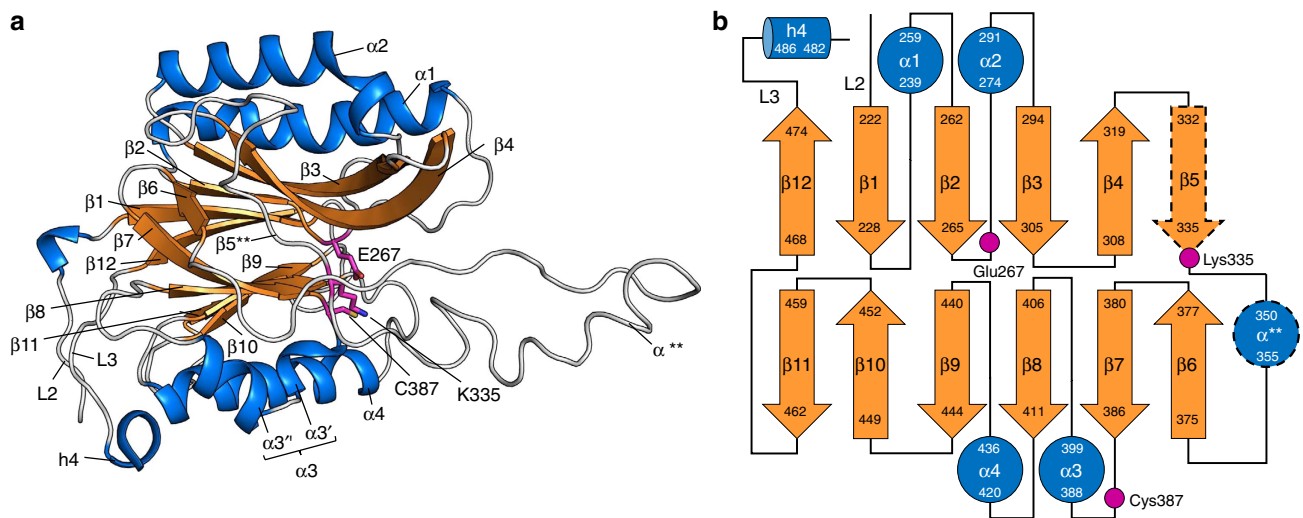

**Figure 6 | Nitrilase-like domain of LntEco. (a)** View from the membrane plane as in Fig. 3a. Colour coded by secondary structure to highlight the αββα sandwich feature of the domain. Catalytic triad residues are shown in stick representation. The asterisks in α** and β5** indicate that the α-helix and β-strand secondary structure elements form in some structures (α-helix found in: LntEco C387A, chain A; LntPae WT. β-strand found in: LntPae WT) but not in others (α-helix absent in: LntEco WT, chains A and B; LntEco C387A, chain B). **(b)** Schematic representation of the secondary structure elements in the nitrilase-like domain. Colour coding follows that used in **a**. Helix α3 consists of two small helices, α3′ and α3″. The dashed lines around α** and β5 indicate that these elements are formed in some structures but not in others as in **a**.

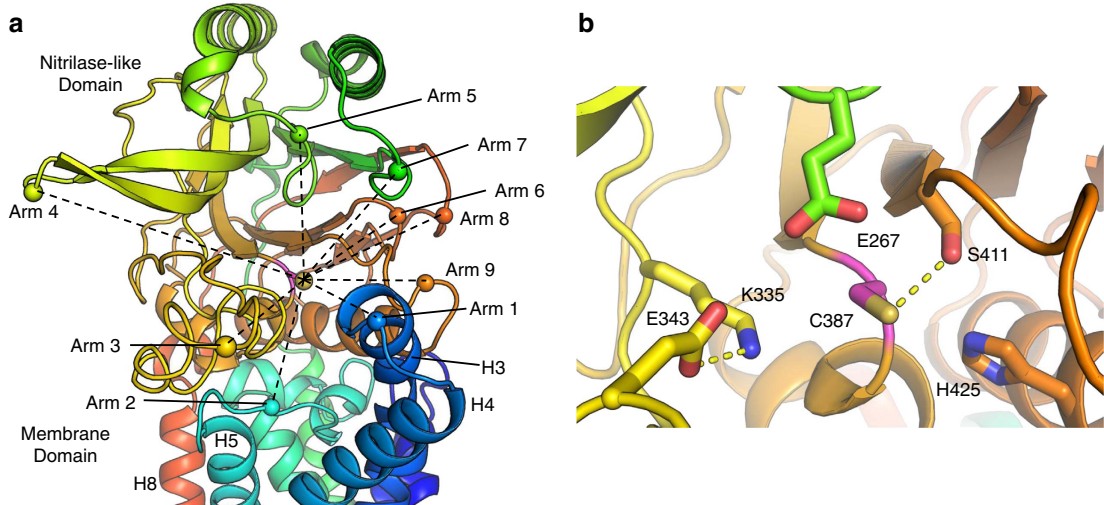

**Figure 7 | Architecture of LntEco active site pocket. (a)** View into the active site pocket from the membrane surface. Helices and loops, referred to as arms, extending from the MD and the ND radiate out creating a funnel-shaped entrance to the active site. Spheres are used to mark the reach of each arm. Reaches are connected to the catalytic Cys387 by dashed lines to communicate a sense of the funnelled nature of the entrance. A description of what constitutes the different arms follows. Arm 1. Periplasmic extensions of H3 and H4 and connecting loop, originating in the MD. Arm 2. L1 between H5 and H6 in the MD. Arm 3. The 40-residue long loop between β5 and β6 linking the two halves of the αββα sandwich in the ND. Arm 4. Long β-strands β3 and β4 and connecting loop in the upper half of the ND sandwich. Arm 5. Loop connecting β1 and α2 in upper half of ND sandwich. Arm 6. Loop between β9 and β10 in bottom half of ND sandwich. Arm 7. Loop between β1 and α1 in upper half of ND sandwich. Arm 8. Loop between β11 and β12, connecting the bottom and top halves of the sandwich. Arm 9. Loop between β8 and α4 in bottom half of the ND sandwich. **(b)** Expanded view into the active site showing the catalytic triad residues, E267, K335 and C387 along with other proximal conserved residues. Side chains are shown in stick representation. The orientation is similar to that in **a**. Dashed lines correspond to distances of ≤ 3.2 Å.

which crystallization occurs. Most lipids decorate the MD of Lnt roughly in a bilayer arrangement reminiscent of lipids in a native membrane (Fig. 8). Some however reside in the H4–H5 cleft and occupy positions between Arm 1 and Arm 3 and above Arm 2 that feed into the active site pocket. In the LntEco-Cys387Ala mutant structure, where lipid density is well defined and plentiful in this region of the protein, the lipids arrange themselves in single file along the hydrophobic pocket entrance into the active site (Fig. 8b,c). The apolar nature of many of the residues in the

periplasmic extensions of H3 and H4 and the three arms is consistent with this finding (Supplementary Fig. 5b, Fig. 8c). These observations suggest that the positions occupied by the structured lipids define the route to the active site taken by the phospholipid and lipoprotein substrates of Lnt. MDS performed with Lnt in a model hydrated membrane, reveal that lipids may enter this region from the bulk membrane (Supplementary Fig. 8). This indicates that there is sufficient space for both substrates to enter the proposed catalytic site one at a time via this

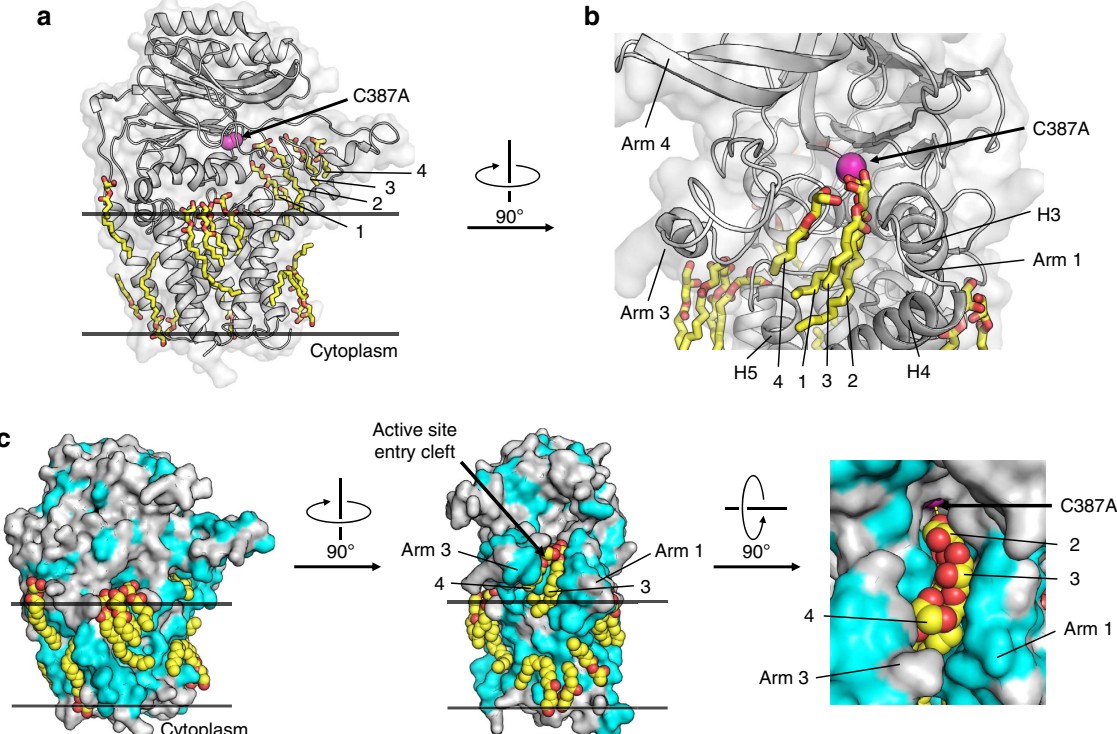

**Figure 8 | Bound lipids define substrate portal in and membrane around Lnt.** (**a**) Structured monoolein lipids in LntEco-C387A decorate the surface of the protein in a manner reminiscent of the membrane bilayer and the portal between the bulk membrane and the enzyme active site. Lipid molecules in the portal line up in single file and are individually numbered from 1 to 4. Lipids are shown in stick representation. Cys387Ala is coloured magenta. (**b**) Expanded view of lipids arranged in single file in the portal facing into the active site and of lipids (shown in stick representation) at the surface of the enzyme. (**c**) Lipid binding to the surface and in the portal of the enzyme. Enzyme shown in surface representation with hydrophobic residues in light blue and polar residues in grey. Orthogonal views presented in left, middle and right panels. An expanded view of lipids (shown as spheres) in the portal is shown in the right panel.

conduit. On the basis of the structured lipids in the crystal structures and simulations it was plausible to dock substrates and products individually into this binding pocket in a physico-chemically reasonable manner and that is consistent with the proposed ping-pong reaction mechanism. Collectively, these observations help explain how the lipid and lipoprotein substrates can effectively migrate out of the bulk membrane along the apolar conduit created by the cleft between H4 and H5 and the space between Arm1 and Arm3 to the active site Cys387 situated some 13 Å above the periplasmic surface of the membrane (Supplementary Fig. 8).

**N-acyltransferase mechanism.** The N-acyltransferase reaction catalysed by Lnt is proposed to take place in two steps (Fig. 9, Supplementary Fig. 9). The first is an acyl transfer reaction where the acyl chain at the *sn*-1 position of the substrate, preferentially PE in the case of Lnt from *E. coli*, is transferred to form a thioester linkage with the γS of the catalytic Cys387 (Fig. 9a–c, Supplementary Fig. 9a–c). In the second step, the acyl chain is transferred from Cys387 to the α-amino group of the dagylated N-terminal cysteine of the lipoprotein (Fig. 9d–f, Supplementary Fig. 9d–f). The reaction follows a ping-pong mechanism where the first product departs the active site before the second substrate enters. The first step is proposed to involve proton abstraction from the γS of Cys387 by catalytic Glu267. This generates a thiolate which, in turn, attacks the ester linkage between the acyl chain and the glycerol backbone of the phospholipid substrate to form a tetrahedral intermediate (Fig. 9b, Supplementary Fig. 9b). A net negative charge on the oxygen attached to the tetrahedral

carbon, the oxyanion, is stabilized by nearby Lys335, and by the proximal oxyanion hole created by the backbone amides of I390, I391 and L392 in α3′ (Fig. 5b). Lys335 has a predicted $pK_a$ some 2.25 units above that for a lysine containing model peptide (Supplementary Table 1). It should be charged under physio-logical conditions for effective oxyanion stabilization. Mutating Lys335 to Ala inactivated the enzyme consistent with the essential role played by this catalytic triad member (Fig. 2d). Glu343 is an invariant residue in the active site pocket. Its side chain carboxyl oxygens are proximal (2.8 Å) to the ε-amino group of Lys335 enhancing its cationic character for oxyanion stabilization. Collapse of the tetrahedral intermediate accompanied by proton abstraction from Glu267 releases the lipid product, lyso-PE. Lyso-PE exits the binding pocket at which point the protein is primed for step 2 which happens as soon as apo-lipoprotein substrate enters and forms the Michaelis complex. The reaction passes through a second proposed tetrahedral intermediate (Fig. 9d, Supplementary Fig. 9e) which forms when the α-amino group at the N terminus of the lipoprotein attacks at and bonds to the carbonyl carbon in the thioester linkage between Cys387 and the acyl chain that is about to undergo transfer. As with the tetrahedral intermediate in the first step of the reaction, the oxyanion is stabilized by Lys335 and the oxyanion hole in the nucleophilic elbow. Collapse of the tetrahedral intermediate gives rise to the mature triacylated lipoprotein product and a reformed enzyme. As soon as the lipoprotein departs and a PE molecule enters the binding pocket, the enzyme is reset for another round of catalysis. Several of the reaction states have been simulated by molecular dynamics and shown to be stable *in silico* (Supplementary Fig. 9, Supplementary Movies 1 and 2).

**Figure 9 | Proposed N-acyltransferase reaction mechanism in Lnt.** Electron lone pairs shown as double dots. (**a**) First Michaelis complex with 1-palmitoyl-2-oleoyl-phosphatidylethanolamine (POPE) substrate. (**b**) First tetrahedral intermediate. Tetrahedral carbon shown in stereochemical representation. The oxyanion bears a negative charge. (**c**) Second Michaelis complex with apo-lipoprotein substrate. (**d**) Second tetrahedral intermediate. (**e**) Product complex. (**f**) Empty active site ready to undergo another reaction cycle. Red curved arrows indicate electron flow. Dashed blue lines denote oxyanion stabilization. GPE, glyceryl-phosphoethanolamine; LP, apo-lipoprotein; DAG, diacylglyceryl; LPE, lyso-PE.

The structure of Lnt is entirely consistent with the proposed mechanism. The predicted $pK_a$ of the $\gamma S$ in the side chain of Cys387 in the Lnt model is 12.7, almost four pH units higher than that of a cysteine containing model peptide (Supplementary Table 1). Therefore, at pH 7.0, Cys387 is fully protonated. However, nearby Glu267 has a side chain $pK_a$ of 7.0 some two and a half pH units above that of Glu in a model peptide. This residue, which is 50% deprotonated at pH 7.0, is in a position to extract a proton from Cys387 thereby generating the highly nucleophilic thiolate. The closest distance between a side chain carboxyl oxygen on Glu267 and the $\gamma S$ of Cys387 is 4.1 Å (3.5 Å in LntPae; Fig. 7b). This is too far for direct interaction and suggests that proton abstraction may involve the mediation of a water molecule to relay transfer from Cys387 to Glu267. Such a water molecule was not seen in the structure perhaps due to resolution limits or disorder. It may also be that flexibility in local backbone and side chain conformation bring the Cys387 and Glu267 into proximity for direct proton exchange. The latter conjecture is supported by MDS, which suggests a spontaneous and consistent hydrogen bond arrangement between the $\gamma S$ proton of Cys387 and the side chain of Glu267 (Supplementary Movie 1). Furthermore, a Glu267Gln mutation inactivated the enzyme consistent with its proposed role in proton abstraction (Fig. 2d). Interestingly, His425 which is highly conserved and located in the active site pocket next to Cys387 (Fig. 7b) has a predicted $pK_a$ of 3.5, some 3 pH units lower than expected (Supplementary Table 1). At pH 7, it is fully deprotonated and should provide additional proton extracting power to increase the nucleophilic character of Cys387. In its protonated state, His425 may also provide stabilization for the oxyanion of the proximal tetrahedral intermediate that forms in each step. In addition to His425, catalytic Cys387 is in hydrogen bonding distance to conserved Ser411 (Fig. 7b). Ser411 is suitably positioned to stabilize the $\gamma S$ of Cys387 as it alternates between thiol and thiolate forms during the two-step transferase reaction.

In several of the crystal structures solved in the course of this work host lipid molecules are visible extending deep into the binding pocket and next to Cys387. In one of these (LntEco-C387A), the $\gamma S$ of Cys387 is 4.5 Å from the hydroxyl on the glycerol backbone of a structured monoolein (Fig. 8). This is compatible with a mechanism where a fatty acyl donor substrate, such as PE, accesses the site with its ester linked acyl chain at the sn-1 position of the glycerol backbone proximal to the nucleophilic thiolate for reaction. The first step in the reaction happens 'spontaneously' yielding an acylated-enzyme intermediate. Indeed, the Lnt enzyme in whole cells has been shown to exist in its acylated form[27]. However, the thioester linkage is labile and can deacylate by hydrolytic cleavage in the absence of second substrate. By contrast, the Cys387Ser mutant has been shown to acylate spontaneously but because the oxygen ester is more stable than the thioester, the subsequent N-acylation step is blocked even in the presence of lipoprotein substrate. We confirmed this in functional assays where the Cys387Ser construct proved to be inactive (Fig. 2d). Efforts aimed at capturing a structure of this intermediate were unsuccessful. In the absence of an actual structure, a model of the palmitoyl-enzyme intermediate has been created and shown to be stable in silico (Supplementary Fig. 10b).

**A structure comparison of LntEco and LntPae.** As noted, LntEco and LntPae have 39% sequence identity. The two structures are remarkably similar with RMSD values over 501 residues of 1.2 Å (Supplementary Fig. 11a). Despite having different residues in the active site pocket, the catalytic triad residues superpose almost exactly on one another (Supplementary Fig. 11c) consistent with the two enzymes catalysing the same N-acyltransferase reaction with similar substrates. Indeed, the phospholipid profile of E. coli and P. aeruginosa membranes are similar with PE and PG together making up 95% of membrane lipids. The rest is mainly cardiolipin. The fatty acids of the phospholipids are predominantly 16 and 18 carbons long. These similarities in membrane lipid composition help rationalize the near structural identity of the two enzymes. Relatedly, LntPae can complement an lnt depletion strain of E. coli and is dominant negative over LntEco at 37 °C (ref. 24). Thus, LntPae is functional in E. coli.

Sequence alignment shows LntEco to have an heptapeptide motif, YSYESAD (Tyr325-Asp331), that is absent in LntPae[28]. The heptapeptide insert forms a loop that clamps into the cleft between α2, β3 and β4 at the top of the ND (Supplementary Fig. 11a,b). Given its location at considerable remove from the

active site, it is not obvious that this difference impacts in any way on catalytic activity or selectivity of the enzyme.

**N-acyltransferase activity**. Detailed characterization of the lipid head group specificity and kinetics of LntEco have identified PE as the preferred substrate ahead of PG. PC was the least favoured acyl donor of the lipids studied[20]. To likewise investigate the head group specificity of LntPae, a lipopeptide band shift assay was performed with LntPae, and with DOPE, DOPG or DOPC as acyl donors. FSL-1-biotin was used as the lipoprotein substrate (Fig. 2). Under these conditions and with PE as the lipid substrate, the reaction ran at the rate of one FSL-1-biotin molecule N-acylated every second. However, transfer rate dropped significantly with PG and PC. Qualitatively therefore, LntPae and LntEco share a similar head group specificity. The structural similarity of the two enzymes is consistent with this finding.

## Discussion

Using substituted cysteine accessibility measurements (SCAM) with whole cells, a membrane topology for LntEco was reported that included six transmembrane helices, a periplasmic ND and an 80 residue long cytoplasmic loop[21]. The latter was proposed to include two hydrophobic helices disposed at the membrane interface. The Ser154Cys mutant used in the SCAM study showed little accessibility to the water-soluble labelling agent leading to the conclusion that Ser154 and the entire cytoplasmic loop were to the cytoplasmic side of the membrane. The current results rationalize this interpretation and show that the two hydrophobic helices referred to in the SCAM based model correspond to H5 and H6 in the crystal structure. Ser154 resides in L1 between H5 and H6 where the MD and ND come together. The arrangement is consistent with the Ser154Cys mutant showing little accessibility.

The crystal structures and MDS suggest that the tips of the arms surrounding the actives site are flexible (Supplementary Fig. 5c,d). Towards the active site, flexibility reduces considerably. In the C387A mutant structure, a row of single monoolein lipid molecules appears corralled in this proposed portal formed by Arm 1, Arm 3 and the H3/H4 cleft that connects the active site pocket with the bulk membrane (Fig. 8). It makes sense therefore that this conduit can be navigated by phospholipid substrate molecules, especially in a sideways orientation with the sn-1 chain on point. On account of its size however, the lipoprotein substrate could experience difficulties with access and egress. For example, if the protein part of the lipoprotein is stably folded all the way to its diacylated N-terminal cysteine it is unlikely to be able to act as a substrate for lack of space. In this folded condition therefore the only way to access the active site would be for the arms to open wide to accommodate the lipoprotein's bulk which can be considerable. This seems unlikely. Such a conformation or range of motion was not observed in the crystal structures nor during MDS. A more reasonable alternative is that the stretch of residues immediately following the N-terminal cysteine of the lipoprotein is unfolded. In this way, it can function as a flexible peptide tether linking the N-terminus, which must access the active site buried in Lnt, and the rest of the lipoprotein. In this scenario, the girth of the tether—the diameter of a peptide chain—would easily be accommodated in the portal to guide the N terminus in its di- and triacylated states into and out of the active site. In support of this proposal, the first 5–6 residues of the lipoprotein subunit CytC are unstructured as observed in the crystal structure of the reaction center complex where all N-terminal residues (C*FEPPPATTTQ) and the lipid modification of CytC are in electron density[29]. Furthermore, sequence analysis (Supplementary

Fig. 12) performed on lipoproteins from E. coli and P. aeruginosa reveal C*SSK(T)P(S)E(K)D(V)S(P)Q(E)P(D)A(S) and C*SSS(L) PPPP(L)PA, respectively, as the most frequently observed residues in the N-terminal undecapeptide. The high frequency of prolines is consistent with an unstructured conformation for the tether, a likely lipoprotein processing motif. Docking and MDS performed with LntEco in complex with the lipodecapeptide substrate FSL-1 (C*GDPKHPKSF) indicates that this mode of access is entirely reasonable (Supplementary Fig. 13).

The acyl chain preferences of LntEco have been investigated[20]. PE with saturated and unsaturated fatty acids at the sn-1 and sn-2 positions, respectively, is an effective substrate. Chain preferences were not investigated in the current study. However, the crystal structures show that the active site is buried at the base of a long channel (Figs 7a and 8). It is there that the acyl chains of the lipid substrates are proposed to reside during reaction. In addition to head group effects therefore, differences in how the acyl chains are accommodated in the confined space of the active site pocket are likely to play a major role in determining lipid substrate selectivity. The same argument can be made regarding the lipid component of the lipoprotein substrate.

Mature lipoproteins are either di- or triacylated. Therefore, some lipoproteins are Lnt substrates while others are not. The origin of the disparity is not known. With lipid substrates, preferences are expressed by Lnt at both head group and acyl chain levels. Likewise for the lipoprotein substrate, preferences based on the protein as well as the lipid component are perhaps to be expected. On the basis of the extant crystal structures of Lnt, it has been proposed that the identity of the acyl chains in the diacylglyceryl moiety of the lipoprotein will impact on lipoprotein substrate selectivity. What about the protein component and to what extent does it influence Lnt lipoprotein preference? Arguments have been presented above regarding the need for an unstructured peptide tether between the lipidated N-terminal cysteine and the folded region of the lipoprotein substrate. This might suggest that provided the tether is unfolded and is long enough, it will suffice as an Lnt substrate. Thus, tether length and structure may determine which lipoproteins will or will not undergo N-acylation.

Downstream of the PTM pathway, triacylated lipoprotein products interact with the Lol trafficking system (Fig. 1)[30]. Residues at the +2 and +3 positions (and +3 and +4 positions in P. aeruginosa), relative to the N-terminal cysteine at position +1, act as sorting signals. Depending on signal identity the lipoprotein either remains in the cytoplasmic membrane or is trafficked to the outer membrane. Presumably, these signature residues matter little to Lnt since, regardless of identity, all of the corresponding lipoproteins must undergo N-acylation. Thus, a degree of insensitivity on the part of Lnt to primary structure in this part of the lipoprotein substrate is expected.

E. coli and P. aeruginosa have between 100 and 200 different types of lipoproteins that range in size from 50 to over 900 residues. The identity of all those that are Lnt substrates is not known. However, it is likely to be considerable. This suggests that the enzyme is promiscuous with regard to protein identity. The fact that biotin C terminally labelled lipopeptide FSL-1 is an Lnt substrate is consistent with this apparent indifference. These observations suggest that Lnt is relatively indifferent to the protein component of its lipoprotein substrate and that the diacylglyceryl moiety on the N-terminal cysteine of a lipoprotein is likely to play a major role in determining whether or not it is an Lnt substrate.

Clearly, the enzymes involved in the synthesis of lipoproteins that play a vital role in the life of the microbial cell are potential targets for antibiotic development programs. Lnt presents a particular challenge because it has a catalytic triad of the nitrilase

type and enzymes, with the same conserved catalytic triad, are found in mammals (Supplementary Fig. 11c)[23]. Fortunately, the substrates of Lnt and mammalian nitrilase-like enzymes are very different. Both substrates of Lnt are fatty acylated molecules while nitrilases cleave carbon-nitrogen bonds in non-lipidated molecules. With high resolution crystal structures of both host and bacterial enzymes, it should be possible to rationally design drugs that selectively target the pathogen.

An additional difficulty arises with developing drugs targeting lipoprotein PTM enzymes that relates to the behaviour of lipoproteins as potent agonists of the immune system. Pathogen recognition and eventual elimination benefits from prompt activation of the innate immune response. Lipoproteins are triggers of this event that involves Toll-like receptors (TLR)[31], TLR1, TLR2 and TLR6. Heterodimerization of TLR2 with TLR1 or TLR6 and lipoprotein complexation sets in train the innate immune responses leading to antigen-specific acquired and long-term immunity. Interestingly, simple lipopeptide mimics of lipoproteins, such as the di- and triacylated peptides, Pam2CSK4 and Pam3CSK4, trigger immune responses suggesting that the N-terminal lipidated cysteine in these ligands is the principal motif stimulating the immune response.

This background highlights the nature of the challenge associated with developing antimicrobial therapies targeting PTM enzymes. The receptors responsible for activating the host innate immune response as well as the enzymes that perform bacterial lipoprotein synthesis both bind lipoproteins. Thus, a ligand found to inhibit lipoprotein synthesis and that is a potential antibiotic may act by binding to the active site of the enzyme mimicking the substrate or product lipoprotein. Likewise, the immune receptors bind lipoproteins tightly and specifically and in a manner that may well resemble that used by the PTM enzymes. In this event, the ligand while possibly a potent antibiotic will have undesirable off-target effects that show up as interfering with and possibly blocking the innate immune response. However, with high-resolution structures of both receptor-lipoprotein and enzyme-lipoprotein complexes and assuming the binding pockets are sufficiently different, it should be possible to rationally design a selective antibiotic without such off-target effects. Crystal structures of TLR-lipopeptide complexes are available[32,33]. Clearly, the need exists for structures of complexes between PTM enzymes and lipoproteins to advance the critical mission of developing safe and selective new antibiotics.

## Methods

**Expression and purification.** The DNA for LntEco and LntPae expression was synthesized and cloned into the pET28a vector using the restriction sites NdeI and XhoI to produce expression constructs with an N-terminal thrombin-cleavable His$_6$-tag (GenScript, USA). For LntEcoC387A, the amino acid substitution was introduced by PCR-based site-directed mutagenesis. Sequences of synthetic genes and primers are included in Supplementary Table 2.

Expression was carried out in chemically competent C43(DE3, Sigma-Aldrich, USA) and C41(DE3, Sigma-Aldrich, USA) cells for LntEco and LntPae, respectively. Cells were transformed with the pET28a vector and grown in kanamycin-supplemented (50 µg ml$^{-1}$) LB agar plates. After overnight incubation, the cells were suspended in 3 × 2 ml of LB and TB for LntEco and LntPae, respectively. 1 ml of this suspension was used to inoculate 1 l of medium. Cultures were grown at 200 r.p.m. and 37 °C to an OD$_{600nm}$ of 1.6 and cooled to 20 °C on ice. Expression of Lnt was induced with 0.5 mM IPTG and cells were grown for 20 h at 20 °C post-induction. Cells were collected at 6,000g for 5 min at 4 °C, resuspended in buffer A (20 mM Tris-HCl, 50 mM NaCl, 0.5 mM EDTA, 1 mM PMSF, pH 8) and lysed at 1,000–1,750 bar using an EmulsiFlex-C5 homogenizer (Avestin, Canada) at 4 °C. Membranes were pelleted by centrifugation at 120,000g for 45 min at 4 °C, solubilized in buffer B (20 mM HEPES-NaOH, 200 mM NaCl, 10%(v/v) glycerol, 1 mM PMSF, 2.5%(w/v) LMNG, pH 7) for 30 min at room temperature (RT) followed by centrifugation at 60,000g for 45 min at 4 °C. The supernatant was supplemented with imidazole to a final concentration of 20 mM and mixed with Ni-NTA Superflow resin (Qiagen, Germany) on a mixer for 60 min at 4 °C. The resin was washed with 200 ml of buffer C (20 mM HEPES-NaOH, 800 mM NaCl, 40 mM imidazole, 10%(v/v) glycerol, 0.5 mM PMSF, 0.05%(w/v)

LMNG, pH 7) and the protein eluted with buffer D (40 mM sodium citrate, 200 mM NaCl, 400 mM imidazole, 10%(v/v) glycerol, 0.5 mM PMSF and 0.1%(w/v) LMNG, pH 6). The protein was further purified using a HiLoad 16/60 Superdex 200 column (GE Healthcare, UK) equilibrated in buffer E (20 mM sodium citrate, 200 mM NaCl, 10%(v/v) glycerol, 0.05%(w/v) LMNG, pH 6). The purified protein was concentrated in a 50 kDa MWCO Amicon Ultra 15 concentrator (Millipore, USA) to ≥13 mg ml$^{-1}$, aliquoted, snap frozen in liquid nitrogen and stored at −80 °C for subsequent use in crystallization and functional assays. For Se-Met labelling of LntEco, B834(DE3) cells were cultured on minimal medium supplemented with 80 mg l$^{-1}$ Se-Met and purified as described above.

**In vitro activity assays.** N-acyl transferase activity of LntPae and LntEco was monitored as a shift in the lipopeptide (fibroblast stimulating ligand; FSL-1-biotin; molecular weight (MW), 2,247 Da) substrate band position on an SDS-PAGE towards higher MW values. Reaction mixtures contained 20-100 nM enzyme, 500 µM phospholipid (POPE, DOPE, DOPG, DOPC; Avanti Polar Lipids, Inc.), 2.5-5 µM FSL-1-biotin (EMC Biochemicals, Germany), 0.1%(w/v) Triton X-100 and either: (i) 50 mM Tris-HCl, 150 mM NaCl, pH 7.2 or (ii) 20 mM sodium citrate, 200 mM NaCl, pH 6.5. Lipids were added from 5%(w/v) stocks in 1%(w/v) Triton X-100. Reactions were run at 37 °C and were stopped by mixing 4 volumes of reaction mixture with 1 volume of 5× SDS loading buffer.

Reaction mixtures containing 25–50 ng of FSL-1-biotin were run on home-made Tris-Tricine PAGE gels supplemented with 6 M urea[34]. The gel consisted of three sequentially polymerized layers: (i) resolving gel, 19 cm: 18% T, 5.25% C, (ii) spacer gel, 2 cm: 11% T, 5.25% C, and (iii) stacking gel, 1.5 cm: 6% T, 5.25% C, where T and C refer, respectively, to total acrylamide and cross-linker concentration in the gel. Gels were run in an SE 660 electrophoresis unit (Amersham Biosciences, UK) at a constant current of 60 mA for ~6 h at RT. FSL-1-biotin bands were transferred electrophoretically onto a nitrocellulose membrane which was subsequently blocked in 30 ml 3%(w/v) bovine serum albumin (BSA) in Tris-buffered saline Tween (TBST) for 10 min, incubated with streptavidin-HRP conjugate (Sigma-Aldrich, USA) at 1:7,500 dilution in 1.5%(w/v) BSA in TBST for 10 min and washed 4 times with 50 ml TBST for 5 min. Blots were developed using a set of SuperSignal solutions (Thermo Scientific, USA). The signal was recorded using a Chemidoc MP gel imaging system (Bio-Rad, USA).

To study the first step in the Lnt catalysed reaction, we developed a simple and robust thin-layer chromatography (TLC)-based functional assay which monitors the formation of a lyso-PE, a product of the reaction catalysed by Lnt. Lyso-PE is formed when Lnt transfers the sn-1-acyl chain from PE to the N-terminal amino group of the apo-lipoprotein. The synthetic lipopeptide FSL-1 was used as the lipoprotein substrate. FSL-1 is a decapeptide based on the N-terminal sequence of a lipoprotein from Mycoplasma pneumoniae. It has been shown previously that the Lnt-acyl intermediate transfers the acyl chain to the α-amino group of the diacylated cysteine at the N terminus of FSL-1 (ref. 20).

18:1-12:0 NBD-PE (NBD-PE, Ex460 Em533) (Avanti Polar Lipids, USA) was used as the substrate acyl donor with which to assay LntEco and LntPae in the presence of FSL-1. NBD-PE is labelled with the fluorophore 7-nitro-2-1,3-benzoxadiazol-4-yl)amino (NBD) at the end of the 12-carbon acyl chain that is in ester linkage at the sn-2 position of glycerol in the PE head group. Lnt specifically transfers the sn-1 chain. Thus, the only NBD-labelled components in the reaction mix should be the NBD-PE substrate and the lyso-NBD-PE product. Both were extracted from the reaction mixture at fixed times post-reaction initiation and separated by TLC. The fluorescence of the NBD-fluorophore was quantified on the TLC plate as detailed below.

Stocks of NBD-PE, solubilized in DMSO (Sigma-Aldrich, USA) at a concentration of 10 mg ml$^{-1}$, were stored at −80 °C. Stocks of FSL-1 (EMC Microcollections, Germany), solubilized in Milli-Q water at a concentration of 5 mg ml$^{-1}$, were stored at −80 °C. Assays were carried out in buffer F (50 mM Tris-HCl pH 7.5, 150 mM NaCl, 1 mM TCEP, 0.02%(w/v) LMNG) at 37 °C.

Time dependence assays were performed in 480 µl of reaction mixture containing 500 µM NBD-PE and 150 µM FSL-1 in buffer F. The reaction was initiated by adding 8 µM LntEco and was carried out at 37 °C with shaking at 180 r.p.m. 40 µl aliquots of the reaction mix were removed at 0, 1, 2, 4, 8, 16, 40, 60, 90 and 120 min after the reaction was started and enzymatic activity was stopped by flash freezing the samples in liquid nitrogen.

The activity of the LntEco mutants Lnt-E267Q, Lnt-K335A and Lnt-C387S was assayed in duplicate in 40 µl reactions containing 520 µM NBD-PE and 160 µM FSL-1 in buffer F. Reactions were initiated by adding 11.6 µM enzyme and were allowed to proceed at 37 °C for 1 h with shaking at 180 r.p.m. The reactions were stopped by flash freezing in liquid nitrogen.

To extract NBD-labelled lipid substrate and product from the reaction mix, 40 µl of 70%(v/v) ethanol were added to the frozen reaction mix samples followed by vortexing for 10–20 s at RT until an homogenous solution was obtained. Lipid substrate and product were extracted by vortexing 30 µl of chloroform with the solution for 45 s. Phase separation was facilitated by centrifugation in a benchtop centrifuge for 2 min at 13,000g and 20 °C. The lower chloroform phase was transferred into a 1.5 ml Eppendorf tube. The tube was left open in a fume hood for 10 min at RT to passively evaporate excess chloroform and to concentrate the lipid. The tube was centrifuged for 2 min at 13,000g and 20 °C and all of the collected organic phase was spotted on a Silica gel 60 F$_{254}$ TLC plate (Merck, USA). The plate was placed in a desiccator at RT under a high vacuum (50 mbar) for 10 min to

remove residual DMSO. Chromatography was carried out with a mobile phase consisting of chloroform:acetone:acetic acid:methanol:water (10:4:2:2:1 by vol). The plate was dried in a stream of nitrogen and imaged using a Biorad Chemidoc MP imager (Fluorescein filter). Fluorescent spots of lyso-NBD-PE were subjected to image analysis using ImageJ[35]. The results were plotted using PRISM 6.0 (GraphPad, USA).

**Crystallization.** For crystallization trials, the Lnt protein was reconstituted into the bilayer of the cubic mesophase following a standard protocol[25,36]. The protein solution was homogenized with monoolein (9.9 MAG) in a coupled syringe mixing device using two volumes of protein solution and three volumes of lipid[37]. Crystallization trials were set up by transferring 50 nl of the protein-laden mesophase onto a silanized 96-well glass sandwich plate followed by 800 nl of precipitant solution using an *in meso* robot[38]. The glass plates were stored in an imager (RockImager 1500, Formulatrix, USA) at 20 °C for crystal growth. Crystals of LntEco were obtained with 8%(v/v) MPD, 0.1 M MES pH 6.0 and 0.4 M ammonium citrate. Crystals of LntPae were obtained with 30%(v/v) PEG-500 DME, 0.1 M sodium citrate pH 5.0 and 0.1 M sodium acetate. Crystals of LntEco and LntPae grew to about $100 \times 70 \times 5 \,\mu m^3$ and about $70 \times 30 \times 5 \,\mu m^3$, respectively, after ~2 weeks. Crystals from the lipid cubic phase were loop-harvested and snap-cooled in liquid nitrogen directly and without added cryo-protectant[39].

**Data collection and processing.** X-ray diffraction experiments were carried out at 100 K on protein crystallography beamlines X06SA-PXI or X10SA-PXII at the Swiss Light Source, Villigen, Switzerland. Measurements were made in steps of 0.1–0.2° at speeds of $1–2°\,s^{-1}$ with either the EIGER 16M or the PILATUS 6M-F detector operated in a continuous/shutterless data collection mode at a sample-to-detector distance of 40-50 cm. For Se-Met SAD phasing, diffraction data were collected on Se-Met-derivative *E. coli* Lnt (LntEco-Se) crystals at the weavelength and flux values of 0.9792 Å and $1.2 \times 10^{11}$ photons per s, respectively. Native data from LntEco crystals were measured at wavelengths and flux values of 1.0 Å and $2.2 \times 10^{11}$ photons per s, respectively. Native data from LntEcoC387A were measured at wavelengths and flux values of 1.0 Å and $6.0 \times 10^{11}$ photons per s. The data sets for LntEco-Se, LntEco and LntEcoC387A were all measured with a $10 \times 30 \,\mu m^2$ X-ray beam size at beamline X10SA-PXII. Native data from LntPae crystals were measured with a $15 \times 40 \,\mu m^2$ X-ray microbeam at wavelengths and flux values of 1.0 Å and $1.2 \times 10^{12}$ photons per s at beamline X06SA-PXI.

Data were processed with XDS[40] and scaled and merged with XSCALE[40]. A 16-fold redundant Se-Met derivative data set to 3.97 Å was obtained by merging the data sets from 3 crystals collected at a wavelength of 0.9792 Å. Data sets for LntEco, LntEcoC387A and LntPae were collected and merged from multiple crystals with 40–150 wedges to 2.9, 2.9 and 3.1 Å resolution, respectively. Data collection parameters are summarized in Table 1.

**Structure determination and refinement.** The SAD method was employed for experimental phasing using a data set from LntEco-Se crystals. Substructure determination was performed with 5,000 SHELXD trials[41] using the HKL2MAP interface[42]. CRANK2 (ref. 43) was used to obtain an initial model, which was completed using the LntEco high resolution data with PHENIX.AutoBuild[44] and manual building in Coot[45]. The structures of LntEcoC387A and LntPae were solved by MR by means of the program MOLREP[46] with LntEco as search model. BUSTER[47] and PHENIX.refine[48] were used during the refinement of all structures. Structure quality scores were obtained using MolProbity implemented from the PHENIX suite[48]. Refinement statistics are reported in Table 1. Figures were generated using PyMOL (http://www.pymol.org)[49].

**Docking and molecular modelling and simulations.** Individual enzyme-ligand complexes were configured and built using initial dockings from Autodock Vina[50] and refined using a combination of Modeller[51], Maestro (Schrödinger Release 2016-4: Maestro, Schrödinger, LLC, New York, NY, 2016) and PyMol (The PyMol Molecular Graphics System, Version 1.8, Schrödinger, LLC), guided by the electron density for the monoolein lipids in the Lnt X-ray structures, the LntEcoC387A construct in particular.

The lipid-modified cysteine parameters were created from lipid parameters for diacylglycerol and palmitoyl moieties and combined with the parameters of the N-terminal cysteines to create the diacylated[17] and triacylated[52] forms of FSL-1.

All MDS were performed using GROMACS v5.1.2 (ref. 53). The Martini 2.2 force field[54] was used to run initial 1 μs Coarse Grained (CG) MD simulations to permit the assembly and equilibration of 1-palmitoyl-2-oleoyl-phosphatidylglycerol (POPG):1-palmitoyl-2-oleoyl-phosphatidylethanolamine (POPE) bilayers around the LntEco and LntPae structures at a 1:3 mole ratio[55]. CG molecular systems were converted to atomistic detail using CG2AT[56], with any unfavourable steric contacts between protein and lipid alleviated using Alchemed[57]. The atomistic systems equate to a total size of ~120,000 atoms and box dimensions in the region of $115 \times 115 \times 115 \,Å^3$. The systems were then equilibrated for 1 ns with the protein restrained before three repeats of 100 ns of unrestrained atomistic MDS, for each configuration of the molecular system (see below), using the Gromos53a6 force field[58].

For both LntEco and LntPae enzymes, simulations were performed with POPE bound to either the protonated or thiolate form of the catalytic cysteine. For the thiolate simulations, the adjacent catalytic glutamate was protonated. Simulations were also run for the palmitoylated form of the cysteine with and without either lyso-PE or diacylated FSL-1 bound. A final set of simulations were carried out with the triacylated FSL-1 product bound. Simulations of the first and second tetrahedral intermediates were also performed. In each case, the bound molecule was positioned based on the monoolein coordinates from the X-ray structures. This equates to a total atomistic simulation time of 2.4 μs per structure. Molecular systems were neutralized with a 150 mM concentration of NaCl.

All simulations were executed at 37 °C, with protein, lipids and solvent separately coupled to an external bath, using the velocity-rescale thermostat[59]. Pressure was maintained at 1 bar, with a semi-isotropic compressibility of $4 \times 10^{-5}$ using the Parrinello-Rahman barostat[60]. All bonds were constrained with the LINCS algorithm[61]. Electrostatics was measured using the Particle Mesh Ewald (PME) method[62], while a cut-off was used for Lennard-Jones parameters, with a Verlet cutoff scheme to permit GPU calculation of non-bonded contacts. Simulations were performed with an integration time step of 2 fs. The MDS were analysed using Gromacs tools, MDAnalysis[63] and locally written python and perl scripts.

Homologous sequences were identified in the UniProt database, using a single iteration of Jackhmmer[64], with the default search parameters. This identified 1,116 non-identical Lnt sequences. Percentage conservation for each residue in the LntEco sequence were mapped to the B-factor column of the LntEco wild-type structure and shown in Supplementary Fig. 6. A Weblogo representation of the sequence alignment was created and shown in Supplementary Figs 6 and 12.

PROPKA[65] was used to estimate the p$K_a$ values of the titratable residues within the Lnt structures (Supplementary Table 1).

**Data availability.** The structures of LntEco WT, LntEco C387A and LntPae WT were deposited into the PDB. The accession codes are 5N6H, 5N6L and 5N6M, respectively. The data that support the findings of this study are available from the corresponding author on reasonable request.

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

## Acknowledgements

We thank past (T. El Arnaout, G. Heinecke, E. Hurley, T. Kwan) and present members of the MS&FB group for their assorted contributions to this study. The assistance and support of beamline scientists at the Swiss Light Source (X06SA and X10SA), Diamond Light Source (I24) and the Advanced Photon Source (23-ID) are acknowledged. The work was funded by Science Foundation Ireland grant 12/IA/1255. M.W. was supported, in part, by Swiss National Science Foundation Early Postdoc. Mobility fellowship grant P2BSP3_15254. D.W. was supported, in part, by the Deutsche Forschungsgemeinschaft (German Research Foundation). P.J.S. was supported by BBSRC grant BB/I019855/1. MDS were performed using the Irish Centre for High-End Computing (ICHEC) facilities.

## Author contributions

M.W. produced and crystallized protein, performed functional assays, collected and processed synchrotron data and performed initial phasing attempts. N.H. refined crystallization conditions for LntPae, produced and crystallized protein and collected synchrotron data. L.V. collected and processed synchrotron data and performed initial phasing attempts. J.B. developed and performed activity assays and collected synchrotron data. C.B. produced and crystallized protein, performed site-directed mutagenesis and collected synchrotron data. D.W. performed biochemical characterization, developed and performed functional assays and collected synchrotron data. C.-Y.H., V.O. and M.Wang collected and processed synchrotron data, and solved and analysed the structure. P.J.S. performed molecular modelling and MDS. N.B. advised on the project. M.C. was responsible for the overall project strategy and management, analysed the structure and wrote the manuscript. Contributions and edits to the written manuscript were provided by co-authors.

## Additional information

**Competing interests:** The authors declare no competing financial interests.

