## [Peer review file · Nature Communications]

Reviewers' comments:

Reviewer #1 (Remarks to the Author):

This paper reports the first crystal structures of the Lnt lipoprotein from *Pseudomonas aeruginosa* and *E. coli*. Based on the atomistic structures, in vitro activity assays and molecular dynamics simulations the authors propose plausible mechanisms by which substrates might interact with the active site. In my opinion this is an interesting and overall scientifically sound work. The manuscript is very well written with clear graphics and the methods are explained in sufficient detail.

In my opinion, this manuscript is clearly of interest for the nature communications community and a wider community in general. Atomic resolution structures of Lnt are a first important step towards rational development of antibiotics.

Reviewer #2 (Remarks to the Author):

Wiktor et al. submitted a detailed description of the solved crystal structure of the lipoprotein N-acyltransferase, an essential membrane protein, highly conserved in Gram-negative bacteria. The unique biochemistry of Lnt has long been under investigation, but any understanding of its enzymatic mechanism

was hampered by a lack of structural data. Most likely, due to the nature of Lnt as an integral membrane protein which presents great technical challenges. These have been overcome by Wiktor et al, whose study makes an important contribution to our current understanding of posttranslational modifications in bacteria and the biogenesis of their membranes.

The experimental background and rationale of the study is straightforward: The authors have overexpressed/synthesized Lnt genes from *E. coli* and *P. aeruginosa*, purified both proteins, raised crystals which allowed them to solve the 3D structures. As both proteins proved to be similar in sequence and revealed nearly identical structures, the authors report mainly results from the *Ec* enzyme. The authors have also included biochemical assays, an inactive version of the enzyme, and molecular dynamics simulations to further conclude on the mode of action. The 10 figures are of an extremely high quality. The same applies to those in the supplementary material. The discussion puts the work in perspective of prior biochemical knowledge as well as the therapeutic potential of Lnt. The manuscript is overall very well written and structured, but far too extensive in many sections. This major point of criticism and several minor points are given below in more detail:

Major:

Several sections of the manuscript are overloaded with narratives, structural details as well as their implications. This dilutes the main message and seems not suitable for a journal which addresses a wide readership with general interest

Some specific examples:

a) The description of the "overall architecture" stretches over more than 1200 words and would probably benefit from two or three further subheadings addressing the MD, ND, and active site architecture separately. It seems as if the figures also support such subdivision

b) The section on the "N-acyltransferase mechanism" also exceeds 1000 words, and could be shortened.

c) It further confuses the reader that the authors added another section on the "N-acyltransferase activity" at the end of the results section. The message that Lnt from *Pae* and *Ec* share a headgroup specificity and that this is supported by their identical structure seems to fit in one sentence.

d) page 16 middle: The authors' conclusions on the recognition of the lipoprotein are

contradictive. Is the enzyme "indifferent" or does it require an unfolded "tether"?

e) The supplementary discussion points into the same direction and is entirely speculative (complex formation, lateral route, etc.) and may or may not convince the true specialists in the field. It should be removed from the manuscript

f) Methods: The description of the in vitro assays is also far too extensive, and should at least be divided to subsections. The same is true for the "docking, molecular modelling ..." paragraph. The authors should address these separately for better readability.

Minor:

Unfortunately, the authors did not give any line numbers.

Introduction:

- The fact that lipoproteins get di- or triacylated depending on growth conditions needs a reference

Results:

- page 3, bottom: The pure proteins WERE functionally...?

- page 3, bottom: A short separate paragraph on the activity would be useful. The last one on the "N-acyltransferase activity" could be deleted instead

- Fig.7: Legend is too extensive, and repeats the text. Instead the amino acids as spheres should be given

-page 7 middle: sitS?

-page 7 middle: ...help poise for reaction the catalytic triad... - This sentence is unclear

-page 7 middle: ...a Thr481Arg mutation inhibited the S-acylation step - Where can this be seen?

-Fig.8: legend, in b) the lipids are shown as sticks not spheres

-page 10 top: "residue" = resides?

-page 10 bottom and Suppl. Fig. 6: colorcoding for H4 and H5 as well as Arm 1 and 2 would be helpful

-page 11 top: Sentence is unclear, what is a "membrane of structures"

-page 11 middle: "proposed mechanism ... takes place"

Discussion

-page 14, 1st paragraph: It is unclear what the authors try to say. According to their structure there is no "cytoplasmic loop"

page 15: The Lyso-lipid part seems entirely speculative and has not been experimentally addressed

Reviewer #3 (Remarks to the Author):

The manuscript by Wiktor et al describes the structures of apolipoprotein N-acyltransferase (Lnt), from *E. coli* and *P. aeruginosa*. Additionally assays have been carried out to analyse the enzymatic activity of the Lnt using a fluorescent based substrate. Finally molecular dynamics simulations were carried out to assess the ability of the enzyme to adopt various conformations as required for acyltransfer chemistry. The manuscript is very well written and enjoyable to read. The structure is very thoroughly described and the results provide important new insights into the function of Lnt. The manuscript has the appropriate scope and technical rigor for publication in *Nature Communications*. I have a number of queries and corrections that should be considered by the authors in a revised version of the manuscript:

Sentence starting on Line 55 doesn't seem to have any logic or reads poorly. It needs to be rewritten to make more sense.

Line 76-78: The explanation of the lipobox sequence situated at the C-terminus is confusing. It can be read that the lipobox sequence is at the C terminus of the protein, not the C terminus of the N-terminal signal sequence. This should be reworded for clarity.

In Supplementary Table 1 the resolution line should line up with the full range of resolution (eg 50 – 3.97) for each of the structures.

When describing the membrane domain some idea of the thickness of the domain in terms of exactly how the membrane edge lines are drawn in Figure 3 should be included. Do the lines denoting the membrane indicate the phosphate portion of the lipids or the hydrophobic portion of the lipids? What is the thickness of the membrane as drawn? On what basis is it decided which parts of the protein are embedded in the membrane and which lie outside of the membrane? Is there a belt of tryptophan residues that define the membrane bound portion of the TM domain?

In order to confirm that pKas of the enzymatic reaction it would be useful to perform the assay under different pH conditions. Have these studies been undertaken? The results suggest the pKas of the residues however confirmation through kinetic activity as a function of pH would be important to confirm these results.

In the interpretation of the mechanism based on the observation of the bound lipid molecules in the binding pocket, the authors state that the first step in the reaction happened "spontaneously" yielding an enzyme intermediate. What is the basis for stating that this happens spontaneously? This would suggest that, in the lipid based environment of the enzyme, it is always active. Is there any evidence for this conclusion? Is there no evidence supporting regulation of the enzyme?

What information is there on lipid substrate specificity? Based on the Lnt activity assay shown in Figure 2b there does appear to be preferred specificity of PE over PG and PC. Are there indications in the structure that provide insights into this specificity?

The authors discuss a possible mechanism of protein product release from Lnt where the final triacylated protein product exits the enzyme using the lyso-lipid from the first step of the reaction as a complex to enhance solubility of the final protein product. This would suggest a mechanism other than ping-pong as the lyso-lipid product from the first reaction would need to be accommodated on the enzyme to enable complexation with the triacylated protein product. This idea has not been adequately developed in the discussion, nor is there any evidence that supports this possibility hence I think it is too overstated to include in the manuscript. Is there any indication in the molecular dynamics simulations that product release may be accompanied by complexation of lyso-lipid? Does the structure provide details as to how the lyso-lipid is bound until complexed with the final product?

Given that Lnt likely exhibits broad substrate specificity for the protein to be acylated, and that the protein substrate must be unwound for it to bind to the Lnt active site, this suggests that Lnt may have an unfolding activity. Does the structure provide any insight into this, particularly in the context of the channel that binds the N terminal portion of the protein substrate? What are the physiochemical characteristics of the channel where the protein substrate is proposed to bind?

The authors use the abbreviate MD to refer to "molecular dynamics" as well as "membrane domain". They should differentiate these two through different abbreviations for clarity .

Response to Reviewers' Comments

Reviewer #1 (Remarks to the Author):

This paper reports the first crystal structures of the Lnt lipoprotein from *Pseudomonas aeruginosa* and *E. coli*. Based on the atomistic structures, in vitro activity assays and molecular dynamics simulations the authors propose plausible mechanisms by which substrates might interact with the active site. In my opinion this is an interesting and overall scientifically sound work. The manuscript is very well written with clear graphics and the methods are explained in sufficient detail.

In my opinion, this manuscript is clearly of interest for the nature communications community and a wider community in general. Atomic resolution structures of Lnt are a first important step towards rational development of antibiotics.

We thank the Reviewer for these complimentary remarks.

Reviewer #2 (Remarks to the Author):

Wiktor et al. submitted a detailed description of the solved crystal structure the apolipoprotein N-acyltransferase, an essential membrane protein, highly conserved in Gram-negative bacteria. The unique biochemistry of Lnt has long been under investigation, but any understanding of its enzymatic mechanism was hampered by a lack of structural data. Most likely, due to the nature of Lnt as an integral membrane protein which present great technical challenges. These have been overcome by Wiktor et al, whose study makes an important contribution to our current understanding of posttranslational modifications in bacteria and the biogenesis of their membranes.

The experimental background and rationale of the study is straight forward: The authors have overexpressed synthesized Lnt genes from *E. coli* and *P. aeruginosa*, purified both proteins, raised crystals which allowed them to solve the 3D structures. As both proteins proved to be similar in sequence and revealed nearly identical structures, the authors report mainly results from the *Ec* enzyme. The authors have also included biochemical assays, an inactive version of the enzyme, and molecular dynamics simulations to further conclude on the mode of action. The 10 figures are of an extremely high quality. The same applies to those in the supplementary material. The discussion puts the work in perspective of prior biochemical knowledge as well as the therapeutic potential of Lnt. The manuscript is overall very well written and structured, but far too extensive in many sections. This major point of criticism and several minor points are given below in more detail:

Major:

Several sections of the manuscript are overloaded with narratives, structural details as well as their implications. This dilutes the main message and seems not suitable for a journal which addresses a wide readership with general interest

Some specific examples:

a) The description of the "overall architecture" stretches over more than 1200 words and would probably benefit from two or three further subheadings addressing the MD, ND, and active site architecture separately. It seems as if the figures also support such subdivision

We agree with the Reviewer that this section might benefit from subheadings. However, we choose the current layout following the style of similar papers published in Nature Communications. Should the Editor indicate that subheadings are needed, we are willing to include them.

b) The section on the "N-acyltransferase mechanism" also exceeds 1000 words, and could be shortened.

We have carefully reviewed this section with a view to shortening it. The language has been tightened up accordingly.

c) It further confuses the reader that the authors added another section on the "N-acyltransferase activity" at the end of the results section. The message that Lnt from *Pae* and *Ec* share a headgroup specificity and that this is supported by their identical structure seems to fit in one sentence.

This is a short paragraph that explains the assay method implemented to quantitatively assess Lnt's N-acyltransferase activity and substrate specificity. Because of its importance we would like to keep it in the revised manuscript.

d) page 16 middle: The authors conclusions on the recognition of the apolipoprotein are contradictive. Is the enzyme "indifferent" or does it require an unfolded "tether"?

We argue that an unfolded stretch of peptide serves to link the dagylated cysteine at the lipoprotein's N-terminus which must access the active site of the enzyme and the bulk of the lipoprotein which presumably is folded. The argument is based on the very limited space available in the portal to accommodate anything other than an unstructured run of amino acids.

e) The supplementary discussion points into the same direction and is entirely speculative (complex formation, lateral route, etc.) and may or may not convince the true specialists in the field. It should be removed from the manuscript

We agree with the Reviewer concerning the speculative nature of this section and have removed the Supplementary Discussion entirely.

f) Methods: The description of the in vitro assays is also far too extensive, and should at least be divided to subsections. The same is true for the "docking,molecular modelling ..." paragraph. The authors should adress these separately for better readability.

These assays are complicated and in one case entirely new. It is in the interests of making them useful to and reproducible in other laboratories that so much detail has been provided. Subheadings of the type referred to by the Reviewer are not used in Nature Communications papers. Should the Editor feel they are needed we will happily add them.

Minor:
Unfortunately, the authors did not give any line numbers.

With apologies. This was not described in the manuscript preparation instructions.

Introduction:
- The fact that lipoproteins get di- or triacylated depending on growth conditions needs a reference

Done.

Results:
- page 3, bottom: The pure proteinS WERE functionally...?

Done.

- page 3, bottom: A short separate paragraph on the activity would be useful. The last one on the ""N-acyltransferase activity" could be deleted instead

Activity is described in considerable detail elsewhere in the manuscript.

- Fig.7: Legend is too extensive, and repeats the text. Instead the amino acids as spheres should be given

Given the importance of the arms that surround and lead into the active site we respectfully request that their full description be retained.

-page 7 middel: sitS?

Done.

-page 7 middle: ...help poise for reaction the catalytic triad... - This sentence is unclear

This sentence has been rewritten for clarity.

-page 7 middle: ...a Thr481Arg mutation inhibited the S-acylation step - Where can this be seen?

The reference to this work has been added. We thank the Reviewer for catching this omission.

-Fig.8: legend, in b) the lipids are shown as sticks not spheres

The correction has been made.

-page 10 top: "residue" = resides?

The correction has been made.

-page 10 bottom and Suppl. Fig. 6: colorcoding for H4 and H5 as well as Arm 1 and 2 would be helpful

We changed the figure as suggested by the Reviewer. However, the resulting figure is more confusing than helpful. Accordingly, we prefer to retain the figure as in the original submission. There are other figures in the manuscript that nicely illustrate lipids in the active site portal.

-page 11 top: Sentence is unclear, what is a "membrane of structures"

The sentence has been rewritten for clarity.

-page 11 middle: "proposed mechanism ... takes place"

The sentence has been rewritten for clarity.

Discussion

-page 14, 1st paragraph: It is unclear what the authors try to say. According to their structure there is no "cytoplasmic loop"

We refer to a short loop in the ND and not to any cytoplasmic loop. It is not clear to what the Reviewer is referring.

page 15: The Lyso-lipid part seems entirely speculative and has not been experimentally addressed

The Reviewer is correct. This is purely speculative but interesting nonetheless. The paragraph has been deleted.

Reviewer #3 (Remarks to the Author):

The manuscript by Wiktor et al describes the structures of apolipoprotein N-acyltransferase (Lnt), from *E. coli* and *P. aeruginosa*. Additionally assays have been carried out to analyse the enzymatic activity of the Lnt using a fluorescent based substrate. Finally molecular dynamics simulations were carried out to assess the ability of the enzyme to adopt various conformations as required for acyltransfer chemistry. The manuscript is very well written and enjoyable to read. The structure is very thoroughly described and the results provide important new insights into the function of Lnt. The manuscript has the appropriate scope and technical rigor for publication in Nature Communications. I have a number of queries and corrections that should be considered by the authors in a revised version of the manuscript: Sentence starting on Line 55 doesn't seem to have any logic or reads poorly. It needs to be rewritten to make more sense.

The sentence has been rewritten for clarity.

Line 76-78: The explanation of the lipobox sequence situated at the C-terminus is confusing. It can be read that the lipobox sequence is at the C terminus of the protein, not the C terminus of the N-terminal signal sequence. This should be reworded for clarity.

The sentence has been rewritten for clarity.

In Supplementary Table 1 the resolution line should line up with the full range of resolution (eg 50 – 3.97) for each of the structures.

The correction has been made.

When describing the membrane domain some idea of the thickness of the domain in terms of exactly how

the membrane edge lines are drawn in Figure 3 should be included. Do the lines denoting the membrane indicate the phosphate portion of the lipids or the hydrophobic portion of the lipids? What is the thickness of the membrane as drawn? On what basis is it decided which parts of the protein are embedded in the membrane and which lie outside of the membrane? Is there a belt of tryptophan residues that define the membrane bound portion of the TM domain?

Indeed, the membrane boundaries are drawn through the phosphate head groups of the lipids in the MDS (see, for example, Supplementary Fig. 6). The thickness of the membrane so defined is 31 Å. This value has now been included in the figure.

LntEco has 16 tryptophans, 12 in the MD and 4 in the ND. The bulk (8/12) of those in the MD decorate the membrane boundaries, as expected. For the most part, those at the boundaries have their indole nitrogen facing out of the membrane toward the aqueous phase.

In order to confirm that pKas of the enzymatic reaction it would be useful to perform the assay under different pH conditions. Have these studies been undertaken? The results suggest the pKas of the residues however confirmation through kinetic activity as a function of pH would be important to confirm these results.

We agree that understanding the pH dependence of the two activities associated with Lnt would be informative. However, such a study would be challenging with the assay methods in place currently. Further, it would require a multitude of control measurements to account for the effect of pH on the protonic equilibrium not only of the enzyme but also of the lipoprotein and phospholipid substrates. For this reason, we consider it beyond the scope of the current work.

In the interpretation of the mechanism based on the observation of the bound lipid molecules in the binding pocket, the authors state that the first step in the reaction happened “spontaneously” yielding an enzyme intermediate. What is the basis for stating that this happens spontaneously? This would suggest that, in the lipid based environment of the enzyme, it is always active. Is there any evidence for this conclusion? Is there no evidence supporting regulation of the enzyme?

There is convincing evidence that Lnt is acylated *in vivo* and this was referred to in the original manuscript (page 13, para 3 original ms, ref 29).

We are not aware of evidence that Lnt is regulated. However, the fact that there are multiple copies of genes for PTM enzymes in bacteria with definite growth development stages is suggestive of some form of regulation. This is referred to in the Introduction (page 2, lines 65-68).

What information is there on lipid substrate specificity? Based on the Lnt activity assay shown in Figure 2b there does appear to be preferred specificity of PE over PG and PC. Are there indications in the structure that provide insights into this specificity?

At the current resolution, there is nothing in the structure to explain lipid substrate specificity. Structures of the enzyme in complex with the different lipid substrates would be hugely informative in this regard; it represents work that is in progress.

The authors discuss a possible mechanism or protein product release from Lnt where the final triacylated protein product exits the enzyme using the lyso-lipid from the first step of the reaction as a complex to enhance solubility of the final protein product. This would suggest a mechanism other than ping-pong as the lyso-lipid product from the first reaction would need to be accommodated on the enzyme to enable complexation with the triacylated protein product. This idea has not been adequately developed in the discussion, nor is there any evidence that supports this possibility hence I think it is too overstated to include in the manuscript. Is there any indication in the molecular dynamics simulations that product release may be accompanied by complexation of lyso-lipid? Does the structure provide details as to how the lyso-lipid is bound until complexed with the final product?

The reviewer is correct. This is pure speculation. It is a really interesting idea that would explain how the final product exits the enzyme and returns to the bulk membrane. However, given that Reviewer 2 also found it on the speculative side, the paragraph has been removed.

Given that Lnt likely exhibits broad substrate specificity for the protein to be acylated, and that the protein substrate must be unwound for it to bind to the Lnt active site, this suggests that Lnt may have an unfolding activity. Does the structure provide any insight into this, particularly in the context of the channel that binds

the N terminal portion of the protein substrate? What are the physiochemical characteristics of the channel where the protein substrate is proposed to bind?

We are not proposing that Lnt unwinds the tether from the lipoprotein substrate to enable its dacylated N-terminal cysteine to access the enzyme's recessed active site. Rather the view is that the lipoprotein comes with an already unstructured N-terminus. Evidence in support of this was referred to in the manuscript that included the presence of a large number of prolines in the putative tether (Supplementary Fig. 9), molecular modelling (Supplementary Fig. 10, Supplementary Movie 2) and a crystal structure that included a lipoprotein with an unfolded N-terminus (ref. 31). Supplementary Fig. 3B shows that the channel is quite hydrophobic. However, the active site and the binding pocket are relatively polar.

The authors use the abbreviate MD to refer to "molecular dynamics" as well as "membrane domain". They should differentiate these two through different abbreviations for clarity.

The relevant changes have been made.